# DC-VideoGen: Efficient Video Generation with Deep Compression Video Autoencoder

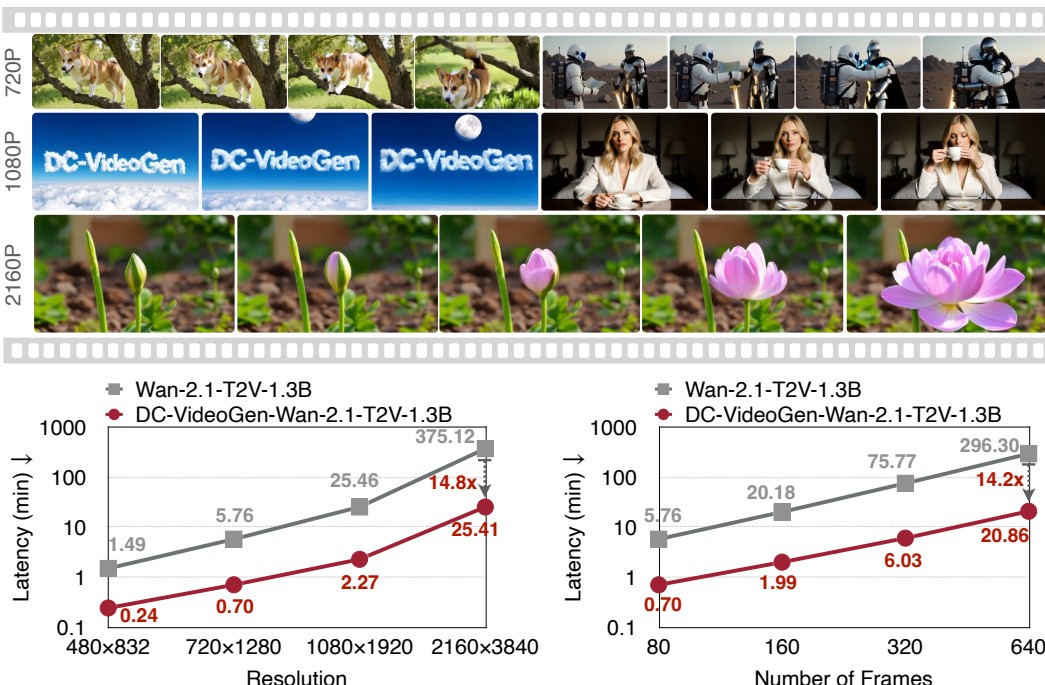

Figure 1: DC-VideoGen can generate high-quality videos on a single NVIDIA H100 GPU with resolutions ranging from 480px, 720px, 1080px, and 2160px. On 2160×3840 resolution, DC-VideoGen delivers 14.8× acceleration compared to the Wan-2.1-T2V-1.3B model.

## Abstract

We introduce DC-VideoGen, a post-training acceleration framework for efficient video generation. DC-VideoGen can be applied to any pre-trained video diffusion model, improving efficiency by adapting it to a deep compression latent space with lightweight fine-tuning. The framework builds on two key innovations: (i) a **Deep Compression Video Autoencoder** with a novel chunk-causal temporal design that achieves 32×/64× spatial and 4× temporal compression while preserving reconstruction quality and generalization to longer videos; and (ii) **AE-Adapt-V**, a robust adaptation strategy that enables rapid and stable transfer of pre-trained models into the new latent space. Adapting the pre-trained Wan-2.1-14B model with DC-VideoGen requires only 10 GPU days on the NVIDIA H100 GPU. The accelerated models achieve up to 14.8× lower inference latency than their base counterparts without compromising quality, and further enable 2160×3840 video generation on a single GPU. We will release both the code and pre-trained models upon publication.

# 1 INTRODUCTION

Video generation has rapidly become a central research focus in generative modeling, driven by its potential to enable applications in creative media, digital communication, virtual product visualization, and world simulation for autonomous driving and robotics. Recent advances in diffusion models and large-scale training have made it possible to synthesize high-quality, temporally coherent videos, substantially narrowing the gap between synthetic and real-world content (Brooks et al., 2024; NVIDIA, 2025). Industry-scale systems such as Veo3 (DeepMind, 2025), Kling (Kuaishou, 2024), Wan (Wan et al., 2025), and Seedance (Gao et al., 2025) have shown that increasing model size and training data leads to significant improvements in video fidelity. Despite these advances, such models remain extremely computationally demanding in both training and inference, limiting their accessibility and practical deployment.

This paper introduces DC-VideoGen, a novel post-training framework for accelerating video diffusion models. Figure 1 showcases high-resolution video samples generated by models accelerated with DC-VideoGen. The framework supports video generation at up to 2160×3840 resolution on a single NVIDIA H100 GPU, achieving a 14.8× inference speedup over the base model. Moreover, DC-VideoGen dramatically reduces training costs compared with training video diffusion models from scratch. For instance, accelerating Wan-2.1-14B (Wan et al., 2025) with DC-VideoGen requires only 10 H100 GPU days — 230× less than the training cost of Wan-2.1-14B (Wan et al., 2025). DC-VideoGen is built upon two core innovations.

**i) Deep Compression Video Autoencoder.** Video data exhibits redundancy across both spatial and temporal dimensions (Xi et al., 2025; Yang et al., 2025b). To mitigate training and inference costs, modern video diffusion models typically employ a video autoencoder that compresses raw videos into a more compact latent space. However, existing video autoencoders generally achieve only moderate compression ratios (e.g., 8× spatial and 4× temporal), which remain insufficient for generating high-resolution or long-duration videos. In Section 3.2, we introduce the Deep Compression Video Autoencoder (**DC-AE-V**), which achieves 32×/64× spatial compression and 4× temporal compression while preserving high reconstruction quality. The core design is a novel chunk-causal temporal modeling approach (Figure 4), which integrates bidirectional information flow within chunks and causal information flow across chunks. This design substantially improves reconstruction quality under deep compression settings (Figure 5) while preserving generalization to longer videos during inference (Figure 3).

**ii) AE-Adapt-V.** After obtaining the deep compression latent space from DC-AE-V, we introduce AE-Adapt-V in Section 3.3.2, which efficiently adapts pre-trained video diffusion models to this latent space through lightweight finetuning (Figure 2). The core design is a video embedding space alignment stage (Figure 7), which helps recover the base model's knowledge and semantics in the new latent space by aligning the patch embedder and output head. These aligned modules provide a robust initialization, enabling rapid recovery of the base model's quality (Figure 6) through LoRA finetuning (Figure 8).

Extensive evaluations on video reconstruction (Table 1), text-to-video generation (Tables 2, 3), and image-to-video generation (Table 4) demonstrate the effectiveness of DC-VideoGen. Across tasks, it consistently provides substantial efficiency gains while achieving comparable or even superior VBench scores. We summarize our main contributions below:

- We introduce DC-VideoGen, a general framework for accelerating video diffusion models. With low-cost post-training finetuning, it delivers substantial efficiency gains in video generation.
- We introduce DC-AE-V, which drastically reduces the number of latent space tokens while preserving high reconstruction quality and generalization to longer videos.
- We introduce AE-Adapt-V, which enables rapid adaptation of pre-trained diffusion models to the latent spaces of new autoencoders.
- DC-VideoGen provides accelerated video diffusion models that preserve the quality of the base models while achieving exceptional efficiency, supporting video generation at up to 2160×3840 resolution on a single GPU. This offers practical advantages for applications requiring efficient video synthesis. Moreover, our accelerated models incur lower fine-tuning and training costs than their base counterparts, enabling faster innovation in the video generation community.

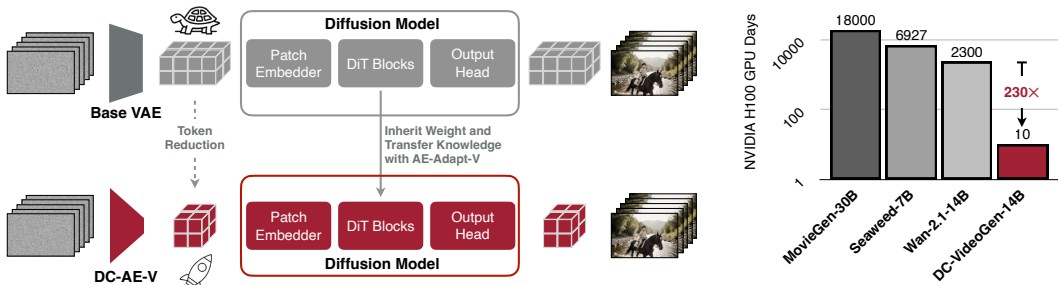

Figure 2: **DC-VideoGen Overview.** DC-VideoGen is a post-training acceleration framework for video diffusion models. It achieves acceleration by transferring models to DC-AE-V's latent space (Section 3.2) and rapidly recovering the base model's quality and semantics using AE-Adapt-V (Section 3.3.2). Compared with training from scratch, DC-VideoGen is far more efficient — for example, DC-VideoGen-14B requires only 10 NVIDIA H100 GPU days, a 230× reduction in training cost relative to Wan-2.1-14B.

## 2 RELATED WORK

**Video Autoencoder.** To circumvent the prohibitive costs of training and running diffusion models in pixel space, latent video diffusion models commonly employ video autoencoders (Blattmann et al., 2023b;a; Zhou et al., 2024; Xu et al., 2024) to compress raw videos into a compact latent space, enabling more efficient generation. A typical configuration in recent works (Zheng et al., 2024; Lin et al., 2024; Zhao et al., 2024a; Chen et al., 2024b; Wang et al., 2024b; Yang et al., 2024; Kong et al., 2024; Wu et al., 2025; Xing et al., 2024; Wan et al., 2025) uses 8× spatial and 4× temporal compression. Some studies (NVIDIA, 2025; Wan et al., 2025; Ma et al., 2025) explore 16× spatial compression to further reduce latent token counts. However, these configurations are often insufficient for high-resolution or long video generation. Inspired by the success of 32× spatial compression in image autoencoders (Chen et al., 2024a), HaCohen et al. (2024); Peng et al. (2025) investigate video autoencoders with a 32× spatial compression ratio, but they suffer from low reconstruction quality or poor generalization to longer videos. In contrast, our DC-AE-V achieves up to 64× spatial compression while maintaining superior reconstruction quality and generalization.

**Efficient Autoencoder Adaptation for Video Diffusion Models.** Closely related to our work, OpenSora 2.0 (Peng et al., 2025) explored adapting pre-trained video diffusion models to their autoencoders by directly loading the pretrained DiT backbone weights while randomly initializing the patch embedder and output head. Their experiments show that this approach produces noticeably blurry videos and fails to match the performance of training from scratch, a finding consistent with our experiments (Figure 6). To address this challenge, we introduce a video embedding space alignment stage that recovers the base model's knowledge in the new latent space.

**Video Diffusion Model Acceleration.** To accelerate video diffusion models, one line of research focuses on reducing the number of diffusion steps (Wang et al., 2023; Lin & Yang, 2024; Li et al., 2024; Zhai et al., 2024; Yin et al., 2024; Wang et al., 2024a; Zhang et al., 2024c; Mao et al., 2025; Lin et al., 2025). Another line explores model compression, including sparsity (Xi et al., 2025; Yang et al., 2025b; Zhang et al., 2025b; Xia et al., 2025; Tan et al., 2025; Li et al., 2025b; Zhang et al., 2025a; Zhao et al., 2025) and quantization (Zhang et al., 2024b;a; Tian et al., 2024; Zhao et al., 2024b; Chen et al., 2025c; Huang et al., 2025; Zhao et al., 2025; Li et al., 2025a). Our DC-VideoGen is complementary to these approaches, as it accelerates video generation by reducing token redundancy.

## 3 METHOD

### 3.1 DC-VIDEOGEN OVERVIEW

Generating high-resolution or long videos with video diffusion models is computationally expensive due to the large number of latent tokens. Furthermore, the prohibitive pre-training costs make developing new video diffusion models both challenging and risky (Gu et al., 2025b).

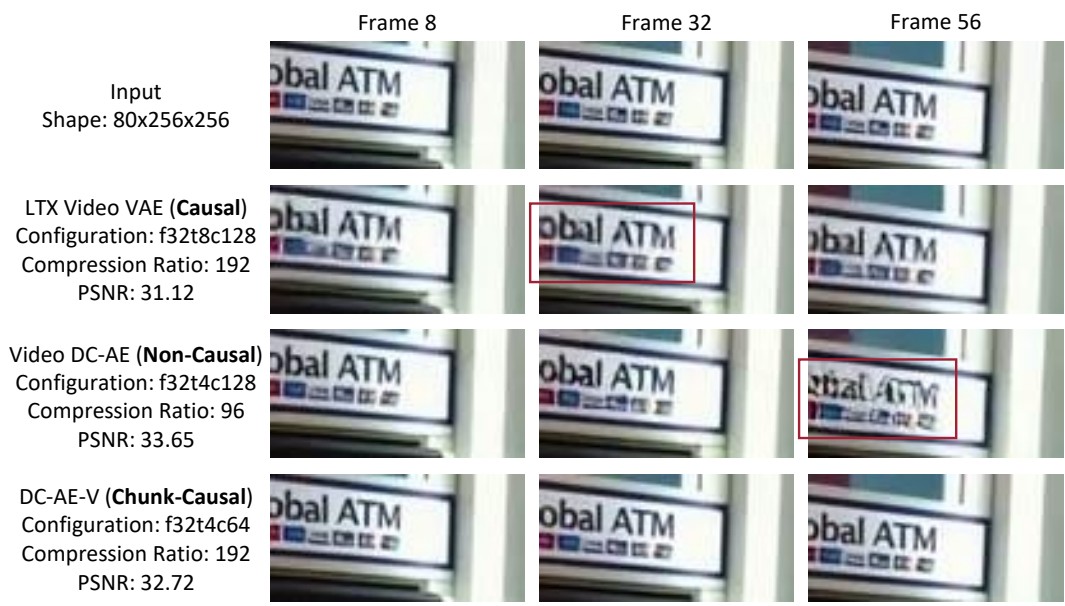

Figure 3: **Video Autoencoder Reconstruction Visualization.** Under deep compression settings, causal video autoencoders suffer from low reconstruction quality. In contrast, non-causal video autoencoders achieve better reconstruction quality but generalize poorly to longer videos.

This paper addresses these challenges from two complementary perspectives (Figure 2, left). First, we drastically reduce the number of tokens using our deep compression video autoencoder. Second, we introduce a cost-efficient post-training strategy to adapt pre-trained models to new autoencoders. This approach substantially lowers the risk, training cost, and reliance on large high-quality datasets. As shown in Figure 2 (right), applying our post-training strategy to Wan-2.1-14B (Wan et al., 2025) takes 10 H100 GPU days — just 0.05% of the training cost of MovieGen-30B (Polyak et al., 2024).

### 3.1.1 PRELIMINARIES AND NOTATION

We use **f**x**t**y**c**z to denote the configuration of a video autoencoder. For example, **f8t4c16** represents a video autoencoder that compresses an input video of size $3 \times T \times H \times W$ into a latent of size $16 \times \frac{T}{4} \times \frac{H}{8} \times \frac{W}{8}$. The compression ratio is defined as

$$\text{Compression Ratio} = \frac{3THW}{c \cdot \frac{T}{t} \cdot \frac{H}{f} \cdot \frac{W}{f}} = \frac{3f^2 t}{c}. \tag{1}$$

Given the same reconstruction quality, a higher compression ratio is generally preferred (Chen et al., 2025b). We refer to diffusion models with an **fxtycz** autoencoder as an **"fxtycz model"**.

A video diffusion model typically comprises a single-layer patch embedder that maps the latent space to the embedding space, transformer blocks, and an output head that projects back to the latent space (Figure 7c). The patch embedder includes a hyperparameter called the patch size $p$, which further spatially compresses the latent by a factor of $p\times$. As shown in (Chen et al., 2024a), for the same total compression ratio, allocating more spatial compression to the autoencoder rather than the patch embedder yields better generation results.

### 3.2 DEEP COMPRESSION VIDEO AUTOENCODER

Existing video autoencoders can be categorized into two groups based on their temporal modeling design: causal and non-causal.

- **Causal.** In causal video autoencoders, information flows only from earlier frames to later frames (Figure 4b). This design naturally supports longer videos during inference, since the encoding and decoding of later frames do not affect earlier ones. However, because each frame can only leverage redundancy from preceding frames, reconstruction accuracy is limited under deep compression settings, as illustrated in Figure 3.

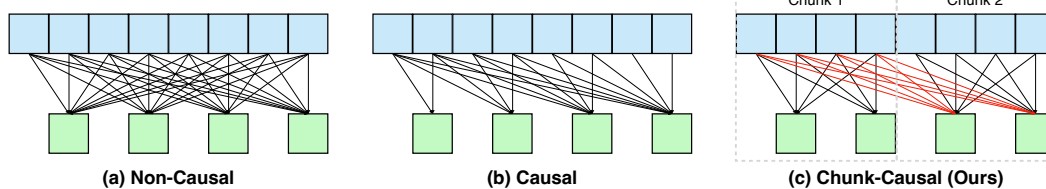

**(a) Non-Causal**    **(b) Causal**    **(c) Chunk-Causal (Ours)**

Figure 4: **Illustration of Chunk-Causal Temporal Modeling in DC-AE-V.** Our chunk-causal temporal modeling preserves causal information flow across chunks while enabling bidirectional flow within each chunk. This design improves reconstruction quality over non-causal temporal modeling, while maintaining generalization to longer videos at inference time.

| Video Autoencoder | Config | Compress. Ratio | Panda70m | | | UCF101 | | | |
|---|---|---|---|---|---|---|---|---|---|
| | | | PSNR↑ | SSIM↑ | LPIPS↓ | PSNR↑ | SSIM↑ | LPIPS↓ | FVD↓ |
| VideoVAEPlus [55] | f8t4c16 | 48 | 36.88 | 0.968 | 0.009 | 35.79 | 0.959 | 0.016 | 2.11 |
| CogVideoX VAE [60] | f8t4c16 | 48 | 35.54 | 0.961 | 0.021 | 34.53 | 0.949 | 0.034 | 8.32 |
| HunyuanVideo VAE [24] | f8t4c16 | 48 | 35.46 | 0.960 | 0.015 | 34.40 | 0.950 | 0.024 | 3.80 |
| IV VAE [52] | f8t4c16 | 48 | 35.32 | 0.959 | 0.017 | 34.84 | 0.955 | 0.025 | 3.71 |
| Wan 2.1 VAE [47] | f8t4c16 | 48 | 34.15 | 0.952 | 0.017 | 33.81 | 0.943 | 0.024 | 3.71 |
| Wan 2.2 VAE [47] | f16t4c48 | 64 | 35.12 | 0.958 | 0.013 | 34.27 | 0.948 | 0.022 | 4.02 |
| StepVideo VAE [32] | f16t8c64 | 96 | 32.17 | 0.930 | 0.043 | 32.17 | 0.930 | 0.043 | 8.23 |
| Video DC-AE$^{\dagger}_{\text{tiling \& blending}}$ [37] | f32t4c128 | 96 | 34.10 | 0.952 | 0.023 | 33.65 | 0.945 | 0.034 | 14.22 |
| Video DC-AE [37] | f32t4c128 | 96 | 31.73 | 0.915 | 0.040 | 31.52 | 0.914 | 0.047 | 26.30 |
| LTX Video VAE [18] | f32t8c128 | 192 | 32.41 | 0.928 | 0.039 | 31.12 | 0.910 | 0.059 | 70.92 |
| DC-AE-V | f32t4c256 | 48 | 39.54 | 0.979 | 0.008 | 37.13 | 0.967 | 0.018 | 2.03 |
| | f32t4c128 | 96 | 37.36 | 0.968 | 0.013 | 34.83 | 0.951 | 0.027 | 5.49 |
| | f32t4c64 | 192 | 35.03 | 0.953 | 0.020 | 32.72 | 0.931 | 0.036 | 13.04 |
| | f32t4c32 | 384 | 33.07 | 0.933 | 0.027 | 30.83 | 0.909 | 0.046 | 29.11 |
| | f64t4c128 | 384 | 32.83 | 0.932 | 0.030 | 30.65 | 0.907 | 0.049 | 32.16 |

Table 1: **Video Autoencoder Reconstruction Results.** $^{\dagger}$Video DC-AE achieves higher PSNR with tiling and blending, but still exhibits poor generalization when applied to longer videos at inference time, as shown in Figure 3.

Building on the causal design, IV-VAE (Wu et al., 2025) notes that, since every $t$ input frames are compressed into a single latent frame, enforcing causality within each group of $t$ frames is unnecessary. To address this, IV-VAE introduces a grouped causal convolution with group size $t$ to improve reconstruction performance. However, the group size is strictly tied to the temporal compression ratio, and as shown in Figure 5, it provides only limited improvements in reconstruction quality over the standard causal design under deep compression settings.

- **Non-Causal.** Non-causal autoencoders allow bidirectional information flow between frames (Figure 4a). This enables each frame to leverage redundancy from both past and future frames, yielding better reconstruction quality under deep compression settings. However, because earlier frames depend on later ones, generalization to longer videos becomes challenging. Techniques such as temporal tiling and blending (Peng et al., 2025) can partially alleviate this issue but often introduce artifacts, including temporal flickering and boundary blurring, as shown in Figure 3.

We introduce a new temporal modeling design, **chunk-causal**, to overcome these limitations (Figure 4c). The key idea is to divide the input video into fixed-size chunks, where the chunk size is treated as an independent hyperparameter. Within each chunk, we apply bidirectional temporal modeling to fully exploit redundancy across frames. Across chunks, however, we enforce causal flow so that the model can effectively generalize to longer videos at inference time. Figure 5 presents the ablation study on chunk size. We observe that increasing the chunk size consistently improves reconstruction quality. In our final design, we adopt a chunk size of 40, as the benefits plateau beyond this point while training costs continue to rise.

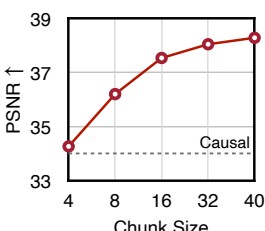

Figure 5: **Ablation Study on Chunk Size.**

**Text-to-Video Generation Results on VBench 480×832**

| Text-to-Video Diffusion Model | Video Autoencoder | Patch Size | Latency (s) ↓ | Overall | Quality | Semantic |
|---|---|---|---|---|---|---|
| | | | | Score ↑ | | |
| | Wan-2.1-VAE-f8t4c16 [47] | 2 | 89.30 | 83.32 | 85.01 | 76.57 |
| Wan-2.1-1.3B [47] | LTX-Video-f32t8c128 [18] | 1 | 6.98 | 83.30 | 85.04 | 76.34 |
| | OpenSora2-f32t4c128 [37] | 1 | 14.57 | 82.27 | 84.53 | 73.24 |
| | Wan-2.2-VAE-f16t4c48 [47] | 2 | 14.59 | 80.38 | 83.20 | 69.08 |
| | DC-AE-V-f64t4c128 | 1 | 3.97 | 83.38 | 85.13 | 76.38 |
| | DC-AE-V-f32t4c32 | 1 | 14.55 | 84.48 | 86.02 | 78.33 |

Table 2: **Comparison of Video Generation Results across Autoencoders.** We adopt the same training setup for all models to ensure apples-to-apples comparisons, i.e., using AE-Adapt-V (Section 3.3.2) to adapt the pretrained Wan-2.1-T2V-1.3B to different autoencoders.

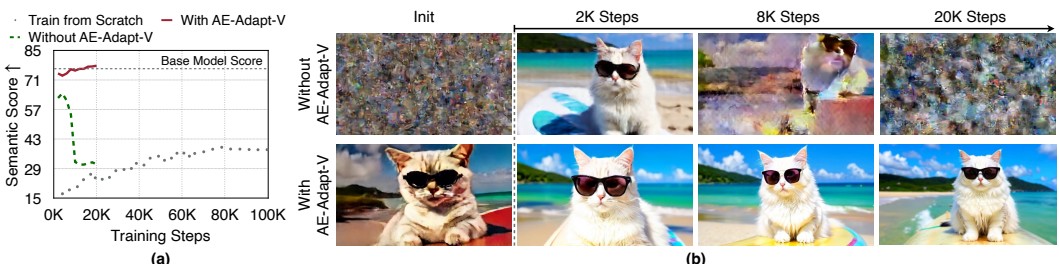

Figure 6: Direct fine-tuning without AE-Adapt-V leads to training instability and suboptimal quality. In contrast, AE-Adapt-V provides a robust initialization that preserves semantics in the new latent space for the video diffusion model, enabling rapid recovery of visual quality and allowing the model to match the base model's performance with lightweight fine-tuning.

**Video Reconstruction Results.** We summarize the comparison between DC-AE-V and prior state-of-the-art video autoencoders in Table 1. Compared with causal video autoencoders such as LTX Video VAE (HaCohen et al., 2024), DC-AE-V achieves higher reconstruction accuracy at the same compression ratio, as well as higher compression ratios for a given accuracy target. Compared with non-causal video autoencoders such as Video DC-AE (Peng et al., 2025), DC-AE-V delivers better reconstruction quality under the same compression ratio while also generalizing better to longer videos (Figure 3).

**Video Generation Results.** In addition to reconstruction performance, we also evaluate DC-AE-V against prior autoencoders on video generation. Table 2 reports ablation results on Wan-2.1-1.3B (Wan et al., 2025), showing that DC-AE-V achieves the best video generation performance. Compared with the base model, DC-AE-V-f64t4c128 provides a 22× speedup while attaining slightly higher VBench scores.

### 3.3 Post-Training Video Autoencoder Adaptation

#### 3.3.1 Naïve Approach, Challenge and Analysis

As discussed in Section 3.1.1, the patch embedder and output head are inherently tied to the latent space representation and thus cannot be transferred when replacing the autoencoder. Consequently, a straightforward approach for adapting pre-trained video diffusion models to new autoencoders is to retain the pre-trained DiT blocks while randomly initializing the patch embedder and output head (Figure 7c, right). This strategy was explored in Peng et al. (2025), where it was found to yield unsatisfactory results.

We evaluate this approach under our settings and observe similar outcomes. As shown in Figure 6a (green dashed line), it fails to match the base model's semantic score. Furthermore, we observe training instability: the model's output degrades to random noise after 20K training steps (Figure 6b,

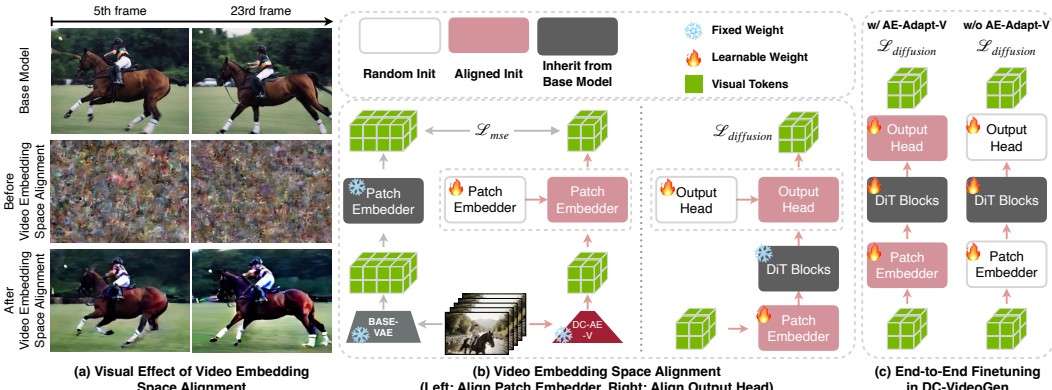

Figure 7: **Illustration of Video Embedding Space Alignment.** We present detailed ablation studies in Figure 9, showing that both alignment steps—patch embedder alignment and output head alignment—are essential for effective video embedding space alignment.

top). We conjecture that this instability arises from the substantial embedding space gap introduced by the new latent space and the randomly initialized patch embedder, which prevents the model from effectively retaining knowledge from the pre-trained DiT weights.

### 3.3.2 OUR SOLUTION: AE-ADAPT-V

To address this challenge, we introduce a video embedding space alignment stage prior to end-to-end finetuning, bridging the gap between embedding spaces and preserving the pre-trained model's knowledge while adapting to a new latent space.

**AE-Adapt-V Stage 1: Video Embedding Space Alignment.**  Figure 7b illustrates the general concept of our video embedding space alignment, where we first align the patch embedder and then align the output head.

For patch embedder alignment, we freeze the base model's patch embedder and train a new patch embedder to map from the new latent space to the embedding space. The objective is to minimize the distance between the base model's embeddings and the embeddings produced by the new patch embedder. Formally, let the embedding of the base model be denoted as $e_b$ with shape $H_b \times W_b \times D$, and the embedding of the new model as $e_n$ with shape $H_n \times W_n \times D$, where $D$ is the embedding channel dimension and $H_n < H_b$, $W_n < W_b$ in our settings. We first spatially downsample $e_b$ using average pooling to match the shape of $e_n$, denoting the result as $e_b'$. The randomly initialized patch embedder is then trained to minimize the following loss function:

$$\mathcal{L} = \text{MSELoss}(e_n, e_b'). \tag{2}$$

With the aligned patch embedder, the output head is then aligned by jointly finetuning it and the patch embedder using the diffusion loss, while keeping the DiT blocks frozen. This process stops once the diffusion loss converges, which takes up to 4K steps in our experiments.

Figure 7a illustrates the visual effect of our video embedding space alignment. Using the aligned patch embedder and output head, we can recover the knowledge and semantics of the base model in the new latent space without updating the DiT blocks. Additional ablation studies are provided in Figure 9, which show that aligning the patch embedder plays the most critical role in video embedding space alignment, while aligning the output head further enhances the quality.

**AE-Adapt-V Stage 2: End-to-End Fine-Tuning with LoRA.**  Video embedding space alignment alone cannot fully match the base model's quality. To close this gap, we perform end-to-end finetuning. Since stage 1 provides a strong initialization, we employ LoRA (Hu et al., 2022) tuning during this stage.

| Text-to-Video Generation Results on VBench 480×832 | | | | | |
|---|---|---|---|---|---|
| Text-to-Video Diffusion Model | Method | Trainable Params (M) | Score ↑ | | |
| | | | Overall | Quality | Semantic |
| Wan-2.1-1.3B [47] | Full-Tune | 1418.90 | 79.81 | 84.02 | 62.98 |
| | LoRA-Tune | 350.37 | **84.48** | **86.02** | **78.33** |

(a) Quantitative Comparison

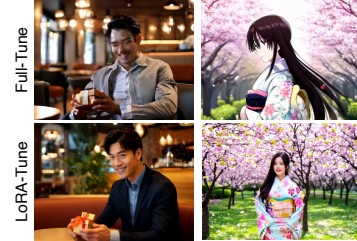

(b) Visual Comparison

Figure 8: **Ablation Study on End-to-End Tuning Strategies.** LoRA attains higher scores than full fine-tuning while requiring far fewer trainable parameters.

Figure 8 compares LoRA tuning with full finetuning. We find that LoRA not only reduces training cost by requiring fewer trainable parameters, but also achieves higher VBench scores and improved visual quality compared with full finetuning. We conjecture that this is because LoRA better preserves the knowledge of the base model.

### 3.4 DC-VIDEOGEN APPLICATION

DC-VideoGen can be applied to any pre-trained video diffusion model. In our experiments, we evaluate it on two representative video generation tasks: text-to-video (T2V) and image-to-video (I2V) generation. We use pre-trained Wan-2.1 models (Wan et al., 2025) as our base models, and denote the resulting accelerated models as DC-VideoGen-Wan-2.1.

The Wan-2.1-I2V models incorporate the image condition by concatenating it with the latent. Since Wan-2.1-VAE and DC-AE-V employ different temporal modeling designs (causal vs. chunk-causal), DC-VideoGen-Wan-2.1 I2V models cannot directly adopt the same approach as Section 3.3.2. To address this, we replicate the given image condition four times and append blank frames to form chunks matching the shape of the video. We then encode these chunks with DC-AE-V and concatenate the resulting features with the latent, which can subsequently be processed in the same manner as in Wan-2.1-I2V.

### 4 EXPERIMENTS

### 4.1 SETUPS

**Implementation Details.** We implement and train all models using PyTorch 2 (Ansel et al., 2024) on 16 NVIDIA H100 GPUs. Three pretrained video diffusion models are employed: Wan-2.1-T2V-1.3B, Wan-2.1-T2V-14B, and Wan-2.1-I2V-14B, each adapted from the original Wan-2.1-VAE to our DC-AE-V. For training, we collected 257K synthetic videos using Wan-2.1-T2V-14B and combined them with 160K high-resolution videos selected from Pexels[1] . Detailed training hyper-parameters are provided in Table 11.

**Efficiency Testbed.** We benchmark the inference latency of all models using TensorRT[2] on a single H100 GPU. For simplicity, we focus exclusively on the transformer backbone, as it constitutes the primary efficiency bottleneck.

**Evaluation Metrics.** Following common practice, we use VBench (Huang et al., 2024) to evaluate text-to-video (T2V) diffusion models and VBench 2.0 (Zheng et al., 2025) for image-to-video (I2V) diffusion models. In addition, we provide visual results generated by our models.

### 4.2 TEXT-TO-VIDEO GENERATION

Table 3 compares DC-VideoGen with leading T2V diffusion models on VBench at 720×1280 resolution. We follow the extended prompt sets provided by the VBench team and conduct all experiments at the same resolution to ensure fair, apples-to-apples comparisons.

---

[1] https://www.pexels.com/videos/
[2] https://github.com/NVIDIA/TensorRT

**Text-to-Video Generation Results on VBench 720×1280**

| Text-to-Video Diffusion Model | Video Autoencoder | #Params (B) | Latency (min) ↓ | Overall | Quality | Semantic |
|---|---|---|---|---|---|---|
| MAGI-1 [43] | - | 4.5 | 21.22 | 79.18 | 82.04 | 67.74 |
| Step-Video [32] | - | 30 | 13.16 | 81.83 | 84.46 | 71.28 |
| CogVideoX1.5 [60] | - | 5 | 6.73 | 82.17 | 82.78 | 79.76 |
| Skyreels-V2 [7] | - | 1.3 | 9.48 | 82.67 | 84.70 | 74.53 |
| HunyuanVideo [24] | - | 13 | 30.35 | 83.24 | 85.09 | 75.82 |
| OpenSora-2.0 [37] | - | 14 | 32.83 | 84.34 | 85.40 | 80.12 |
| Wan-2.1-1.3B[†] [47] | Wan-2.1-VAE-f8t4c16 | 1.3 | 5.76 | 83.38 | 85.67 | 74.22 |
| DC-VideoGen-Wan-2.1-1.3B | DC-AE-V-f32t4c32 | 1.3 | 0.70 | 84.63 | 86.67 | 76.48 |
| Wan-2.1-14B [47] | Wan-2.1-VAE-f8t4c16 | 14 | 27.52 | 83.73 | 85.77 | 75.58 |
| DC-VideoGen-Wan-2.1-14B | DC-AE-V-f32t4c32 | 14 | 3.58 | 84.83 | 86.80 | 76.93 |

Table 3: **Results on Text-to-Video Generation**. [†]Native Wan-2.1-T2V-1.3B is limited to 480×832 resolution, so we fine-tune it on our dataset to support 720×1280 generation.

**Image-to-Video Generation Results on VBench 720×1280**

| Image-to-Video Diffusion Model | Video Autoencoder | #Params (B) | Latency (min) ↓ | Overall | Quality | I2V |
|---|---|---|---|---|---|---|
| CogVideoX-5b-I2V [60] | - | 5 | 6.72 | 86.70 | 78.61 | 94.79 |
| HunyuanVideo-I2V [24] | - | 13 | 30.39 | 86.82 | 78.54 | 95.10 |
| Step-Video-TI2V [32] | - | 30 | 13.18 | 88.36 | 81.22 | 95.50 |
| MAGI-1 [43] | - | 4.5 | 21.25 | 89.28 | 82.44 | 96.12 |
| Wan-2.1-14B [47] | Wan-2.1-VAE-f8t4c16 | 14 | 27.88 | 86.86 | 80.83 | 92.90 |
| DC-VideoGen-Wan-2.1-14B | DC-AE-V-f32t4c32 | 14 | 3.67 | 87.73 | 81.39 | 94.08 |

Table 4: **Results on Image-to-Video Generation.**

Compared with the base Wan-2.1 models, DC-VideoGen-Wan-2.1 achieves higher scores while being significantly more efficient. For example, DC-VideoGen-Wan-2.1-14B reduces latency by 7.7× and improves the VBench score from 83.73 to 84.83 relative to Wan-2.1-14B. Compared with other T2V diffusion models, DC-VideoGen-Wan-2.1 achieves both the highest VBench score and the lowest latency. Video samples can be found in Figure 13 and the webpage in supplementary material.

## 4.3 IMAGE-TO-VIDEO GENERATION

Table 4 reports our results on VBench I2V at 720×1280 resolution. Consistent with the T2V findings, DC-VideoGen-Wan-2.1-14B outperforms the base Wan-2.1-14B by achieving a higher VBench score while reducing latency by 7.6×.

Compared with other I2V diffusion models, DC-VideoGen-Wan-2.1-14B provides highly competitive results with exceptional efficiency, running 5.8× faster than MAGI-1 and 8.3× faster than HunyuanVideo-I2V. Video samples can be found in Figure 12 and the webpage in supplementary material.

## 4.4 ABLATION OF AE-ADAPT-V

In AE-Adapt-V, we introduce video embedding space alignment training to initialize the newly added patch embedder and output head. To verify the effectiveness of video embedding space alignment in AE-Adapt-V, we introduce two baseline initialization strategies:

- Baseline 1: Randomly initialize the patch embedder and output head.
- Baseline 2: Freeze the DiT backbone and directly fine-tune the patch embedder and output head with diffusion loss.

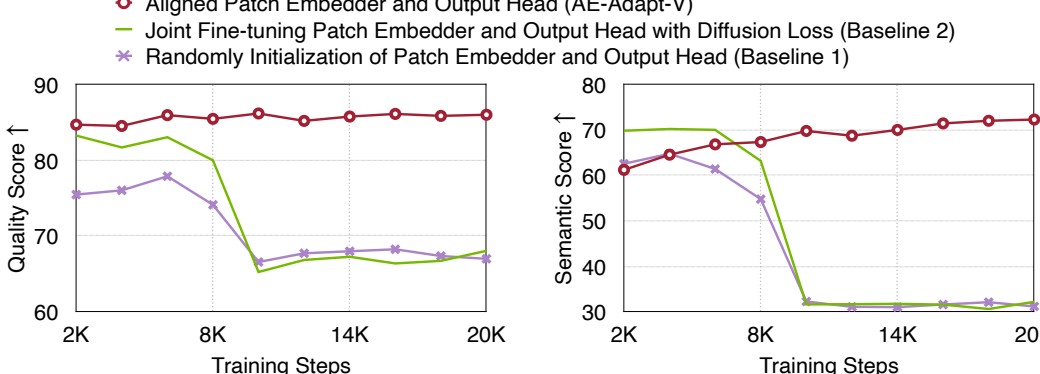

Figure 9: **Ablation Study on Video Embedding Space Alignment in AE-Adapt-V.** Compared to the two baselines, our AE-Adapt-V demonstrates stable training and superior performance during autoencoder adaptation.

| Human Evaluation Results on 720P Image-to-Video Generation | | | | | |
|---|---|---|---|---|---|
| **Visual Quality** | | | **Text Alignment** | | |
| **DC-VideoGen-Wan-2.1-I2V-14B is better** | **Wan-2.1-I2V-14B is better** | **Equal** | **DC-VideoGen-Wan-2.1-I2V-14B is better** | **Wan-2.1-I2V–14B is better** | **Equal** |
| **56.7%** | 34.3% | 9.0% | **58.8%** | 32.9% | 8.3% |

Table 5: **Human Evaluation Results.** We conduct a human evaluation on Amazon Mechanical Turk. DC-VideoGen-Wan-2.1-I2V-14B achieves a higher win rate than Wan-2.1-I2V-14B in both visual quality and text alignment.

Using these initialization strategies, we then perform end-to-end LoRA fine-tuning. As shown in Fig. 9, the Video Embedding Space Alignment in AE-Adapt-V is an essential component for ensuring stability when changing the latent space.

## 4.5 HUMAN EVALUATION

We conducted a human evaluation comparing Wan-2.1-I2V-14B and DC-VideoGen-Wan-2.1-I2V-14B on 720P video generation. To ensure an unbiased assessment, we used Amazon Mechanical Turk, a crowdsourcing platform. Our evaluation covered 116 testing prompts, with 5 raters assigned to each prompt, resulting in a total of 580 questionnaires.

For each prompt, we first generated a 720P image using FLUX.1-Krea, then synthesized a 720P video conditioned on that image. The human-evaluation interface is shown in Fig. 14. Participants were asked two questions: which video has better visual quality (including appearance and motion), and which video aligns more closely with the prompt. The results are summarized in Table. 5. DC-VideoGen-Wan-2.1-I2V-14B demonstrates superior visual quality and text alignment compared to Wan-2.1-I2V-14B, achieving a higher win rate.

## 5 CONCLUSION

We introduce DC-VideoGen, a post-training framework that accelerates video diffusion models by combining a deep compression video autoencoder with an efficient adaptation strategy. DC-VideoGen achieves up to $14.8\times$ faster inference and drastically reduces training costs, while preserving or even improving video quality. These findings highlight that efficiency and fidelity in video generation can advance together, making large-scale video synthesis more practical and accessible for both research and real-world applications. We further discuss potential extensions and current limitations of our framework in Section A.7.

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

# A  APPENDIX

## A.1  THE USE OF LLM

The use of LLMs was strictly limited to the final manuscript writing stage, specifically for the purpose of correcting grammatical errors and polishing the text.

## A.2  ADDITIONAL DETAILS OF DC-AE-V

### A.2.1  MODEL ARCHITECTURE

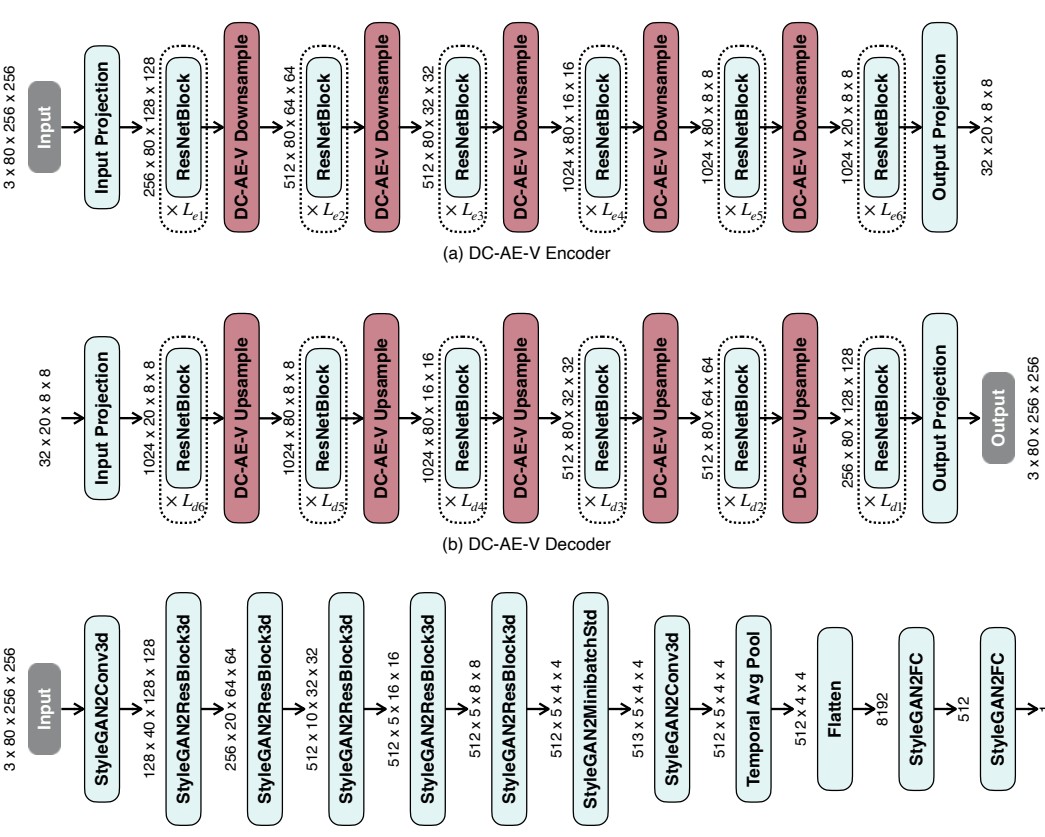

Figure 10: **Model Architecture of DC-AE-V.**

Figure 10 presents the detailed model architecture of an f32t4c32 DC-AE-V. Both the encoder and decoder are composed of six stages, each built from 3D ResNet (He et al., 2016) blocks. The first five stages perform only spatial downsampling and upsampling, while the final stage handles temporal downsampling and upsampling. Following DC-AE (Chen et al., 2024a), we incorporate Residual Autoencoding to facilitate optimization during downsampling and upsampling. For adversarial training, we extend the StyleGAN2 discriminator (Karras et al., 2020) to process video inputs.

Algorithm 1 presents the implementation details of our chunk-causal design. We mainly modify the temporal padding of the 3D convolution to enable chunk causal modeling. Specifically, for all chunks, the right temporal padding is set to zero to ensure causality between chunks, as shown in Algorithm 1 (line 2). For chunks other than the first chunk, their left temporal padding is set to the last $\frac{k_t - 1}{2}$ frames of the previous chunk to gather information, as shown in Algorithm 1 (line 4).

---

**Algorithm 1 Chunk-Causal Implementation.**

---

**Require:** Chunk size $T$, input activation $X \in \mathbb{R}^{c_{\text{in}} \times nT \times h \times w}$, convolution kernel $W \in \mathbb{R}^{c_{\text{out}} \times c_{\text{in}} \times k_t \times k_h \times k_w}$

1: Set $X \leftarrow X.\text{reshape}(c_{\text{in}}, n, T, h, w).\text{transpose}(0, 1) \in \mathbb{R}^{n \times c_{\text{in}} \times T \times h \times w}$

2: Set $\text{pad}_{t,\text{right}} \leftarrow \mathbf{0} \in \mathbb{R}^{n \times c_{\text{in}} \times \frac{k_t - 1}{2} \times h \times w}$

3: Set $\text{pad}_{t,\text{left,first}} \leftarrow \mathbf{0} \in \mathbb{R}^{1 \times c_{\text{in}} \times \frac{k_t - 1}{2} \times h \times w}$

4: Set $\text{pad}_{t,\text{left,subsequent}} \leftarrow x[:-1, :, -\frac{k_t - 1}{2} :] \in \mathbb{R}^{(n-1) \times c_{\text{in}} \times \frac{k_t - 1}{2} \times h \times w}$

5: Set $\text{pad}_{t,\text{left}} \leftarrow \text{concatenate}([\text{pad}_{t,\text{left,first}}, \text{pad}_{t,\text{left,subsequent}}], \dim = 0) \in \mathbb{R}^{n \times c_{\text{in}} \times \frac{k_t - 1}{2} \times h \times w}$

6: Set $X \leftarrow \text{concatenate}([\text{pad}_{t,\text{left}}, X, \text{pad}_{t,\text{right}}], \dim = 2) \in \mathbb{R}^{n \times c_{\text{in}} \times (T + k_t - 1) \times h \times w}$

7: Set $X \leftarrow \text{spatial\_pad}(X, k_h - 1, k_w - 1) \in \mathbb{R}^{n \times c_{\text{in}} \times (T + k_t - 1) \times (h + k_h - 1) \times (w + k_w - 1)}$

8: Set $X \leftarrow \text{convolution}(X, W) \in \mathbb{R}^{n \times c_{\text{out}} \times T \times h \times w}$

9: Set $X \leftarrow X.\text{transpose}(0, 1).\text{reshape}(c_{\text{out}}, nT, h, w) \in \mathbb{R}^{c_{\text{out}} \times nT \times h \times w}$

10: **return** $X$

---

| Video Autoencoder | Config | Compress. Ratio | Panda70m PSNR ↑ | SSIM ↑ | LPIPS ↓ | UCF101 PSNR ↑ | SSIM ↑ | LPIPS ↓ | FVD ↓ | ActivityNet PSNR ↑ | SSIM ↑ | LPIPS ↓ | FVD ↓ | Kinetics600 PSNR ↑ | SSIM ↑ | LPIPS ↓ | FVD ↓ |
|---|---|---|---|---|---|---|---|---|---|---|---|---|---|---|---|---|---|
| VideoVAEPlus [55] | f8t4c16 | 48 | 36.88 | 0.968 | 0.009 | 35.79 | 0.959 | 0.016 | 2.11 | 35.68 | 0.955 | 0.016 | 0.96 | 36.73 | 0.960 | 0.014 | 0.93 |
| CogVideoX VAE [60] | f8t4c16 | 48 | 35.54 | 0.961 | 0.021 | 34.53 | 0.949 | 0.034 | 8.32 | 34.47 | 0.946 | 0.034 | 5.16 | 35.40 | 0.951 | 0.032 | 4.18 |
| HunyuanVideo VAE [24] | f8t4c16 | 48 | 35.46 | 0.960 | 0.015 | 34.40 | 0.950 | 0.024 | 3.80 | 34.41 | 0.943 | 0.024 | 3.16 | 35.40 | 0.950 | 0.022 | 2.59 |
| IV VAE [52] | f8t4c16 | 48 | 35.32 | 0.959 | 0.017 | 34.84 | 0.955 | 0.025 | 3.71 | 35.03 | 0.948 | 0.025 | 1.88 | 36.27 | 0.956 | 0.022 | 1.52 |
| Wan 2.1 VAE [47] | f8t4c16 | 48 | 34.15 | 0.952 | 0.017 | 33.81 | 0.943 | 0.024 | 3.71 | 33.82 | 0.938 | 0.025 | 1.76 | 35.04 | 0.946 | 0.022 | 1.49 |
| Wan 2.2 VAE [47] | f16t4c48 | 64 | 35.12 | 0.958 | 0.013 | 34.27 | 0.948 | 0.022 | 4.02 | 34.41 | 0.943 | 0.021 | 1.56 | 35.57 | 0.950 | 0.019 | 1.51 |
| StepVideo VAE [32] | f16t8c64 | 96 | 32.17 | 0.930 | 0.043 | 32.17 | 0.930 | 0.043 | 8.23 | 32.08 | 0.922 | 0.047 | 5.29 | 33.02 | 0.931 | 0.044 | 4.62 |
| Video DC-AE$_{\text{tiling \& blending}}$ [37] | f32t4c128 | 96 | 34.10 | 0.952 | 0.023 | 33.65 | 0.945 | 0.034 | 14.22 | 33.55 | 0.938 | 0.033 | 7.92 | 34.73 | 0.946 | 0.030 | 6.81 |
| Video DC-AE [37] | f32t4c128 | 96 | 31.73 | 0.915 | 0.040 | 31.52 | 0.914 | 0.047 | 26.30 | 31.34 | 0.901 | 0.049 | 17.52 | 32.39 | 0.915 | 0.044 | 15.43 |
| LTX Video VAE [18] | f32t8c128 | 192 | 32.41 | 0.928 | 0.039 | 31.12 | 0.910 | 0.059 | 70.92 | 31.29 | 0.900 | 0.058 | 45.51 | 32.26 | 0.911 | 0.056 | 42.30 |
| DC-AE-V | f32t4c256 | 48 | 39.54 | 0.979 | 0.008 | 37.13 | 0.967 | 0.018 | 2.03 | 37.28 | 0.965 | 0.016 | 0.94 | 38.11 | 0.969 | 0.016 | 0.88 |
| | f32t4c128 | 96 | 37.36 | 0.968 | 0.013 | 34.83 | 0.951 | 0.027 | 5.49 | 35.06 | 0.948 | 0.025 | 2.59 | 35.91 | 0.953 | 0.024 | 2.47 |
| | f32t4c64 | 192 | 35.03 | 0.953 | 0.020 | 32.72 | 0.931 | 0.036 | 13.04 | 33.02 | 0.927 | 0.035 | 6.22 | 33.87 | 0.935 | 0.033 | 6.24 |
| | f32t4c32 | 384 | 33.07 | 0.933 | 0.027 | 30.83 | 0.909 | 0.046 | 29.11 | 31.08 | 0.901 | 0.045 | 13.83 | 32.01 | 0.912 | 0.042 | 13.05 |
| | f64t4c128 | 384 | 32.83 | 0.932 | 0.030 | 30.65 | 0.907 | 0.049 | 32.16 | 30.90 | 0.899 | 0.049 | 15.63 | 31.77 | 0.910 | 0.047 | 15.50 |

Table 6: **Additional Video Reconstruction Results.**

### A.2.2 DATASET

Our DC-AE-V is trained on a mixture of video and image datasets. The video datasets include subsets of Panda70m (Chen et al., 2024c) and OpenVid1m (Nan et al., 2024). The image datasets include ImageNet21k (Ridnik et al., 2021), Mapillary Vistas (Neuhold et al., 2017), DataComp (Gadre et al., 2023), WiderFace (Yang et al., 2016), WiderPerson (Zhang et al., 2019), TextCaps (Sidorov et al., 2020), and Unsplash (Unsplash, 2020).

### A.2.3 EVALUATION

We evaluate all video autoencoders on $80 \times 256 \times 256$ videos using PSNR, SSIM (Wang et al., 2004), LPIPS (Zhang et al., 2018), and FVD (Unterthiner et al., 2018). The evaluation set includes 1,000 unseen videos from Panda70m (Chen et al., 2024c), 3,783 test videos from UCF101 (Soomro et al., 2012), 5,044 test videos from ActivityNet 1.3 (Caba Heilbron et al., 2015), and the first 5,000 test videos from Kinetics600 (Carreira et al., 2018).

For VideoVAEPlus (Xing et al., 2024), we use the '16z' version. For CogVideoX VAE (Yang et al., 2024), we use the model from THUDM/CogVideoX-2b. For HunyuanVideo VAE (Kong et al., 2024), we use the model from hunyuanvideo-community/HunyuanVideo. For IV VAE (Wu et al., 2025), we use the 'ivvae_z16_dim96' version. For Wan 2.1 VAE (Wan et al., 2025), we use the model from Wan-AI/Wan2.1-T2V-14B-Diffusers, and for Wan 2.2 VAE (Wan et al., 2025), we use the model from Wan-AI/Wan2.2-TI2V-5B. For StepVideo VAE (Ma et al., 2025), we use the 'vae_v2' from stepfun-ai/stepvideo-t2v. For Video DC-AE (Peng et al., 2025), we use the model from hpcai-tech/Open-Sora-v2-Video-DC-AE. Finally, for LTX Video VAE (HaCohen et al., 2024), we use the model from Lightricks/LTX-Video-0.9.7-dev. When an autoencoder cannot process 80-frame videos, we pad extra frames at the end and exclude them from the reconstructions when computing evaluation metrics.

| Video Autoencoder | Config | Compress. Ratio | Panda70m | | | UCF101 | | | | ActivityNet | | | | Kinetics600 | | | |
|---|---|---|---|---|---|---|---|---|---|---|---|---|---|---|---|---|---|
| | | | PSNR ↑ | SSIM ↑ | LPIPS ↓ | PSNR ↑ | SSIM ↑ | LPIPS ↓ | FVD ↓ | PSNR ↑ | SSIM ↑ | LPIPS ↓ | FVD ↓ | PSNR ↑ | SSIM ↑ | LPIPS ↓ | FVD ↓ |
| VideoVAEPlus [55] | f8t4c16 | 48 | 36.66 | 0.967 | 0.009 | 35.18 | 0.955 | 0.018 | 2.73 | 35.59 | 0.954 | 0.016 | 1.16 | 36.58 | 0.959 | 0.015 | 1.09 |
| CogVideoX VAE [60] | f8t4c16 | 48 | 35.42 | 0.960 | 0.021 | 34.28 | 0.946 | 0.036 | 9.85 | 34.48 | 0.946 | 0.035 | 6.05 | 35.36 | 0.951 | 0.033 | 4.78 |
| HunyuanVideo VAE [24] | f8t4c16 | 48 | 35.55 | 0.960 | 0.015 | 34.32 | 0.949 | 0.025 | 3.99 | 34.65 | 0.945 | 0.023 | 2.82 | 35.63 | 0.951 | 0.021 | 2.40 |
| IV VAE [52] | f8t4c16 | 48 | 35.29 | 0.959 | 0.017 | 34.66 | 0.953 | 0.026 | 4.38 | 35.07 | 0.949 | 0.025 | 2.18 | 36.26 | 0.956 | 0.022 | 1.77 |
| Wan 2.1 VAE [47] | f8t4c16 | 48 | 34.41 | 0.954 | 0.016 | 33.69 | 0.942 | 0.025 | 4.46 | 34.01 | 0.940 | 0.023 | 2.09 | 35.15 | 0.947 | 0.022 | 1.81 |
| Wan 2.2 VAE [47] | f16t4c48 | 64 | 35.61 | 0.961 | 0.012 | 34.10 | 0.947 | 0.023 | 4.94 | 34.67 | 0.945 | 0.020 | 1.85 | 35.73 | 0.952 | 0.018 | 1.74 |
| StepVideo VAE [32] | f16t8c64 | 96 | 33.13 | 0.943 | 0.031 | 31.82 | 0.926 | 0.046 | 11.65 | 32.15 | 0.922 | 0.046 | 6.13 | 33.02 | 0.931 | 0.044 | 5.55 |
| Video DC-AE$_{tiling \& blending}$ [37] | f32t4c128 | 96 | 34.15 | 0.952 | 0.023 | 33.21 | 0.941 | 0.036 | 16.98 | 33.63 | 0.938 | 0.033 | 8.40 | 34.72 | 0.946 | 0.030 | 7.13 |
| Video DC-AE [37] | f32t4c128 | 96 | 35.55 | 0.961 | 0.017 | 33.62 | 0.943 | 0.034 | 17.63 | 34.24 | 0.943 | 0.029 | 8.98 | 35.12 | 0.949 | 0.027 | 7.63 |
| LTX Video VAE [18] | f32t8c128 | 192 | 32.26 | 0.927 | 0.040 | 30.78 | 0.906 | 0.062 | 80.15 | 31.24 | 0.899 | 0.058 | 45.86 | 32.13 | 0.910 | 0.057 | 41.41 |
| DC-AE-V | f32t4c256 | 48 | 39.44 | 0.978 | 0.008 | 36.32 | 0.964 | 0.020 | 2.45 | 37.30 | 0.965 | 0.016 | 1.03 | 38.04 | 0.968 | 0.016 | 0.95 |
| | f32t4c128 | 96 | 37.19 | 0.968 | 0.013 | 34.09 | 0.946 | 0.029 | 6.41 | 35.03 | 0.948 | 0.025 | 2.74 | 35.79 | 0.953 | 0.024 | 2.54 |
| | f32t4c64 | 192 | 34.83 | 0.952 | 0.020 | 32.05 | 0.924 | 0.039 | 14.72 | 32.87 | 0.925 | 0.034 | 6.18 | 33.64 | 0.932 | 0.033 | 5.97 |
| | f32t4c32 | 384 | 32.94 | 0.932 | 0.028 | 30.35 | 0.903 | 0.049 | 35.46 | 31.06 | 0.901 | 0.045 | 15.29 | 31.90 | 0.911 | 0.043 | 14.21 |
| | f64t4c128 | 384 | 32.63 | 0.930 | 0.030 | 30.10 | 0.900 | 0.052 | 36.78 | 30.81 | 0.897 | 0.048 | 15.14 | 31.60 | 0.908 | 0.046 | 14.10 |

Table 7: **Additional Video Reconstruction Results on 40-Frame Videos.**

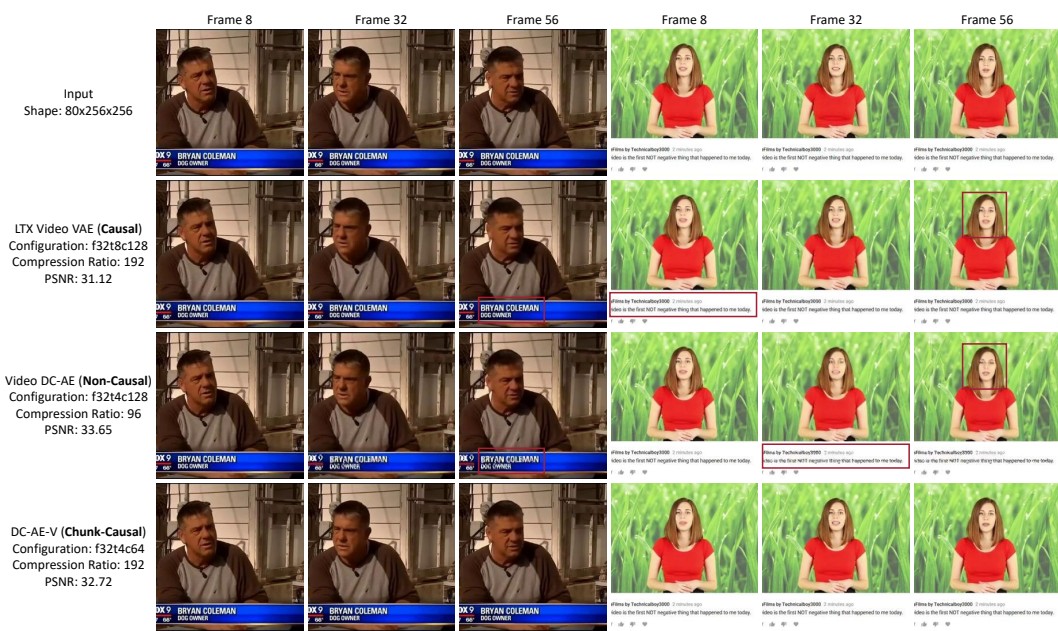

Figure 11: **Additional Video Autoencoder Reconstruction Visualization.** Under deep compression settings, causal video autoencoders suffer from low reconstruction quality. In contrast, non-causal video autoencoders achieve better reconstruction quality but generalize poorly to longer videos.

### A.2.4 ADDITIONAL RECONSTRUCTION RESULTS

Table 6 presents the full reconstruction results. Our DC-AE-V consistently achieves superior accuracy and generalizes effectively to longer videos across a range of benchmarks.

Table 7 presents reconstruction results on 40-frame videos. In this setting, our chunk-causal design degenerates to the non-causal design, yet our DC-AE-V still consistently delivers superior reconstruction accuracy. DC-AE-V is trained with chunk size 40, while previous VideoVAEPlus (Xing et al., 2024) and Video DC-AE (Peng et al., 2025) are trained on 16 and 32 frames, respectively. Our reconstruction result benefits from the larger chunk size, which is consistent with the trends shown in Figure 5.

Figure 11 presents additional reconstruction examples. Our DC-AE-V demonstrates superior reconstruction accuracy and generalization ability to longer videos, especially for small texts and human faces.

| Video Autoencoder | Config | Compress. Ratio | #Params (M) Total | Encoder | Decoder | Latency (s) Encode | Decode |
|---|---|---|---|---|---|---|---|
| VideoVAEPlus [55] | f8t4c16 | 48 | 490.35 | 176.17 | 240.55 | 0.67 | 1.12 |
| CogVideoX VAE [60] | f8t4c16 | 48 | 215.58 | 92.22 | 123.37 | 0.38 | 0.89 |
| HunyuanVideo VAE [24] | f8t4c16 | 48 | 246.48 | 100.34 | 146.14 | 0.46 | 0.81 |
| IV VAE [52] | f8t4c16 | 48 | 241.85 | 101.70 | 140.15 | 0.55 | 0.84 |
| Wan 2.1 VAE [47] | f8t4c16 | 48 | 126.89 | 53.59 | 73.29 | 0.27 | 0.43 |
| Wan 2.2 VAE [47] | f16t4c48 | 64 | 704.69 | 149.63 | 555.05 | 0.15 | 0.43 |
| StepVideo VAE [32] | f16t8c64 | 96 | 498.82 | 109.89 | 388.93 | 0.23 | 0.69 |
| Video DC-AE$_{\text{tiling \& blending}}$ [37] | f32t4c128 | 96 | 459.46 | 220.44 | 239.02 | 0.34 | 0.52 |
| Video DC-AE [37] | f32t4c128 | 96 | 459.46 | 220.44 | 239.02 | 0.45 | 0.69 |
| LTX Video VAE [18] | f32t8c128 | 192 | 1246.91 | 694.10 | 552.81 | 0.05 | 0.04 |
| DC-AE-V | f32t4c256 | 48 | 2138.64 | 916.66 | 1221.98 | 0.34 | 0.36 |
| | f32t4c128 | 96 | 2138.64 | 916.66 | 1221.98 | 0.34 | 0.36 |
| | f32t4c64 | 192 | 2138.64 | 916.66 | 1221.98 | 0.34 | 0.36 |
| | f32t4c32 | 384 | 2138.64 | 916.66 | 1221.98 | 0.34 | 0.36 |
| | f64t4c128 | 384 | 2717.28 | 1152.89 | 1564.39 | 0.34 | 0.36 |

Table 8: **Video Autoencoder Efficiency.** The encoding and decoding latency is measured for a $80 \times 256 \times 256$ video using an NVIDIA H100 GPU.

**Text-to-Video Generation Results on VBench 720×1280**

| Models | Temporal Flickering | Subject Consistency | Motion Smoothness | Dynamic Degree | Aesthetic Quality | Imaging Quality | Background Consistency | Overall Consistency |
|---|---|---|---|---|---|---|---|---|
| Wan-2.1-T2V-1.3B | 99.15 | 94.97 | **98.36** | 67.78 | 70.20 | 68.44 | 97.99 | **25.97** |
| DC-VideoGen-Wan-2.1-T2V-1.3B | **99.18** | **96.58** | 98.34 | **72.78** | **72.00** | **68.72** | **98.00** | 25.41 |

| Models | Object Class | Multiple Objects | Human Action | Color | Spatial Relationship | Scene | Appearance Style | Temporal Style |
|---|---|---|---|---|---|---|---|---|
| Wan-2.1-T2V-1.3B | **89.11** | 72.07 | 94.05 | 81.54 | 65.90 | 44.83 | **21.61** | **23.22** |
| DC-VideoGen-Wan-2.1-T2V-1.3B | 88.73 | **75.98** | **94.64** | **89.16** | **78.20** | **44.86** | 21.20 | 22.97 |

Table 9: **Detailed Results on VBench.**

(a) Latency (minutes per video)

| Models | Resolution 480×832 | 720×1280 | 1080×1920 | 2160×3840 |
|---|---|---|---|---|
| Wan-2.1-1.3B [47] | 1.49 | 5.76 | 25.46 | 375.12 |
| DC-VideoGen-Wan-2.1-1.3B | 0.24 | 0.70 | 2.27 | 25.41 |
| Speedup | 6.2× | 8.2× | 11.2 × | 14.8× |

(b) Latency (minutes per video)

| Models | Number of Frames 80 | 160 | 320 | 640 |
|---|---|---|---|---|
| Wan-2.1-1.3B [47] | 5.76 | 20.18 | 75.77 | 296.30 |
| DC-VideoGen-Wan-2.1-1.3B | 0.70 | 1.99 | 6.03 | 20.86 |
| Speedup | 8.2× | 10.1× | 12.6× | 14.2× |

Table 10: **Detailed Efficiency Benchmark Results.**

### A.2.5 EFFICIENCY

Table 8 presents an efficiency comparison for various video autoencoders. The encoding and decoding latency is measured for a $80 \times 256 \times 256$ video using an Nvidia H100 GPU. Our DC-AE-V achieves comparable latency compared with previous video autoencoders.

## A.3 DETAILED EVALUATION RESULTS ON VBENCH

Table 9 reports detailed metrics on VBench. DC-VideoGen-Wan-2.1-T2V-1.3B outperforms the base Wan-2.1-T2V-1.3B on 11 of the 16 metrics.

## A.4 DETAILED EFFICIENCY BENCHMARK RESULTS

Table 10 presents detailed efficiency results measured on an NVIDIA H100 GPU.

## A.5 DETAILED TRAINING HYPERPARAMETERS OF AE-ADAPT-V

Table 11 lists the detailed hyperparameters for AE-Adapt-V.

| Training Stage | Hyperparameter | Value |
|---|---|---|
| Patch Embedder Alignment | learning rate | 1e-4 |
| | warmup steps | 0 |
| | batch size | 4 |
| | training steps | 20k |
| | optimizer | AdamW, betas=[0.9, 0.999] |
| Output Head Alignment | learning rate | 1e-4 |
| | warmup steps | 0 |
| | batch size | 32 |
| | training steps | 4k (Wan-2.1-1.3B) / 3k (Wan-2.1-14B) |
| | optimizer | AdamW, betas=[0.9, 0.999] |
| End-to-End Fine-Tuning | learning rate | 5e-5 |
| | warmup steps | 1k |
| | training steps | 20k (Wan-2.1-1.3B) / 6k (Wan-2.1-14B) |
| | batch size | 32 |
| | optimizer | AdamW, betas=[0.9, 0.999] |
| | weight decay | 1e-3 |
| | LoRA (rank, alpha) | (256, 512) |
| Resolution Increasing | 480px→720px, training steps | 1000 |
| | 720px→1080px, training steps | 500 |
| | 1080px→2160px, training steps | 200 |

Table 11: **Training Hyperparameters of AE-Adapt-V.**

## A.6 QUALITATIVE COMPARISON WITH THE PRE-TRAINED MODELS

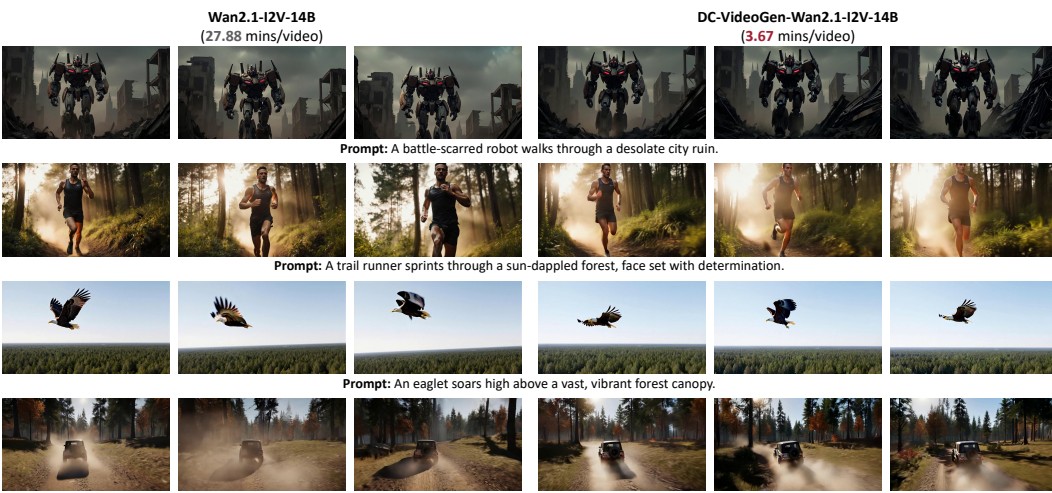

Figure 12: **Visual Comparison of DC-VideoGen-Wan2.1-I2V-14B and the Base Model Wan2.1-I2V-14B.**

Figure 12 and Figure 13 provide a qualitative comparison between DC-VideoGen and the base models. We observe that DC-VideoGen-Wan2.1-I2V-14B and DC-VideoGen-Wan2.1-T2V-14B retain the generation quality of Wan2.1-I2V-14B and Wan2.1-T2V-14B while reducing the latency by around 87%.

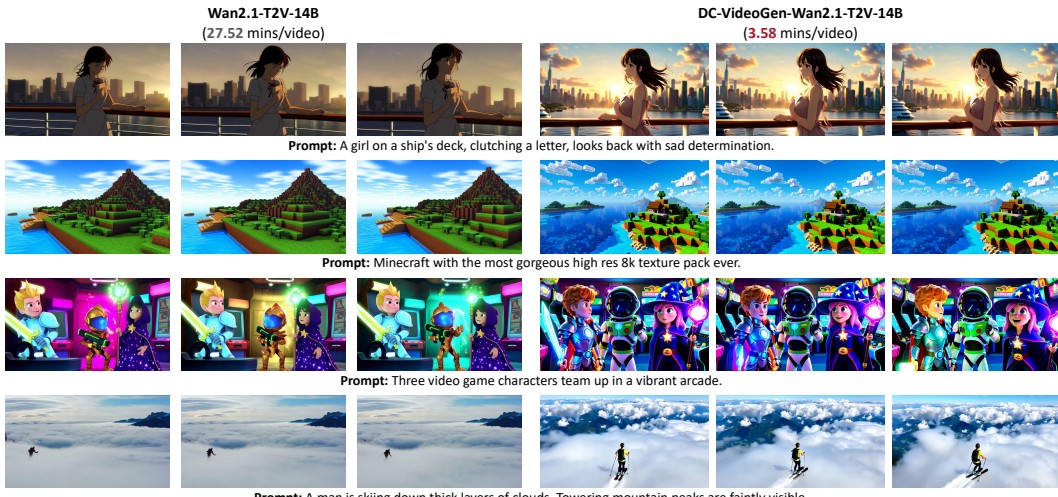

Figure 13: **Visual Comparison of DC-VideoGen-Wan2.1-T2V-14B and the Base Model Wan2.1-T2V-14B.**

### A.7 LIMITATION AND FUTURE WORK

**Limitations.** As a post-training framework, DC-VideoGen accelerates video diffusion models through lightweight fine-tuning, effectively leveraging the rich knowledge encoded in the pre-trained model. Consequently, its performance is strongly dependent on the quality of the pre-trained model.

**Future Work.** DC-VideoGen substantially reduces the training and inference costs of video diffusion models, especially when scaling to higher resolutions. For the next step, we plan to extend our framework for long video generation, leveraging techniques from (Gu et al., 2025a; Yang et al., 2025a).

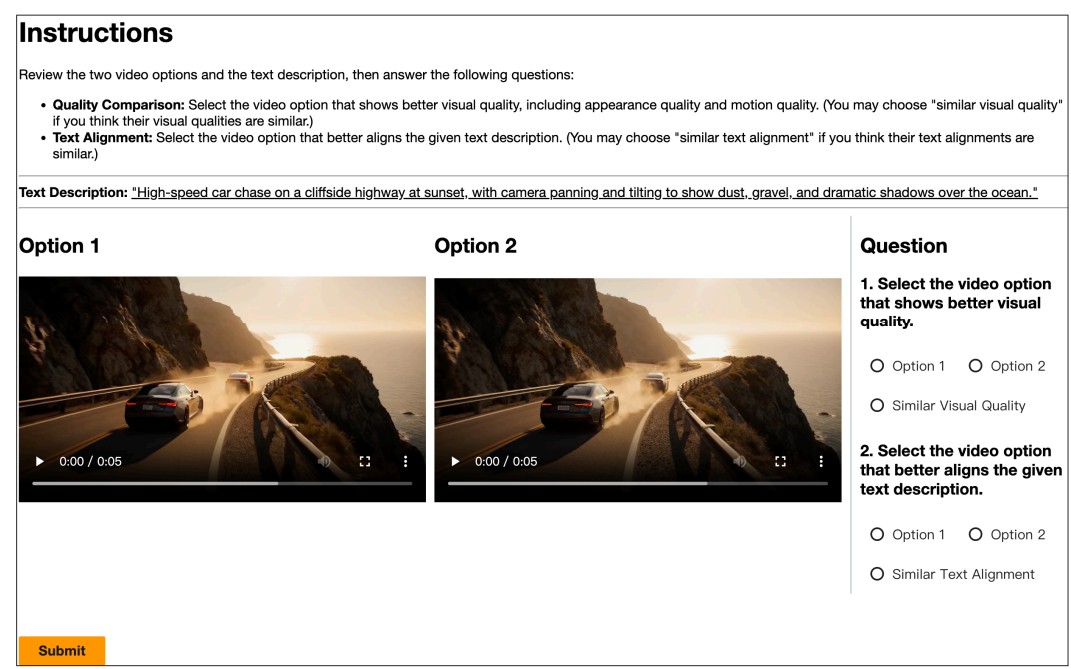

Figure 14: **Illustration of the Human Evaluation Interface.** We use Amazon Mechanical Turk to distribute questionnaires for assessing the human preference.