# OpenReview forum: "DC-VideoGen: Efficient Video Generation with Deep Compression Video Autoencoder"
_ICLR.cc/2026/Conference — Submitted to ICLR 2026_

### Official Review · Reviewer_3xJk · 2025-10-31

**Soundness:** 2
**Presentation:** 3
**Contribution:** 3
**Rating:** 4
**Confidence:** 5

**Summary:**

The paper's central thesis is that existing large-scale video diffusion models can be migrated to an extremely compressed latent space via a post-training approach, achieving over an order of magnitude in inference acceleration with negligible loss in visual quality. The authors propose a two-step framework to this end. First, they design a new, highly compressive video autoencoder (DC-AE-V) to create this compact latent representation. Second, they introduce a specialized fine-tuning procedure (AE-Adapt-V) that is claimed to "seamlessly" adapt a pre-trained large model to this new, smaller latent space without causing model collapse. The paper asserts that the proposed method achieves state-of-the-art results in both reconstruction quality and generation performance.

**Strengths:**

1. The authors' commitment to open-sourcing their work is commendable.
2. The provision of an HTML file in the supplementary materials for direct and intuitive visual comparison is a welcome feature.
3. The paper's reconstruction and generation results are highly impressive.
4. The proposed method is straightforward and easy to follow.

**Weaknesses:**

1. Unlike magvit-v2, the authors do not treat the first frame of a video differently. I'm curious whether the paper's method can handle the image frame effectively and how its image reconstruction performance compares to Hunyuan VAE. Additionally, I'd like to know how the authors trained the images under the chunk causal format.
2. The chunk-causal VAE proposed by this method significantly outperforms the non-causal VideoDC-AE, which appears counterintuitive. Since VideoDC-AE employs temporal slicing during testing to ensure the number of tested frames is less than or equal to the number of training frames. Additionally, it overlaps different slices, for example, by 8 frames, to ensure subsequent slices have sufficient access to historical frames. Therefore, conceptually, the proposed approach and VideoDC-AE's method are fundamentally similar. The primary difference lies in the mechanism for passing history, latent feature passing versus overlapping input frames. The paper's chunk-causal method occupies a middle ground between IV-VAE and OpenSora2, offering limited innovation.
3. The paper's AE-Adapt-V appears to be over-designed. What is the necessity of introducing the MSE loss? Why not simply freeze the DiT backbone and fine-tune the embedding and output heads simultaneously in a single stage? This more straightforward and sensible baseline is conspicuously absent from the paper's comparative experiments.
4. It is well-known that VBench scores can be gamed by fine-tuning on carefully curated datasets. Therefore, I place more emphasis on perceptual quality than the reported metrics. The qualitative comparisons in the appendix are unconvincing. Aside from exhibiting greater motion, the generations from DC-VideoGen-Wan-2.1-14B appear inferior to the original Wan 2.1 in terms of texture and aesthetics. This is particularly concerning given that the provided cases are few and likely cherry-picked.

**Questions:**

See Weaknesses.

---

> ### Author Response · Authors · 2025-11-24
>
> > _**Q1: DC-AE-V on Images**_
>
> To handle the first frame in our chunk-casual design, we replicate the image four times and treat the replicates as a four-frame video. This four-frame video is encoded to a one-frame latent representation and then decoded back into a four-frame video. We only keep the first frame of the decoded video as the image reconstruction result. In this manner, we can apply joint image-video training to maintain the image reconstruction ability. As shown in the table below, our DC-AE-V achieves image reconstruction results comparable to other video autoencoders.
>
> | ImageNet 256      |rFID|PSNR |SSIM |LPIPS|
> | ----------------- |----|-----|-----|-----|
> | VideoVAEPlus      |1.10|28.30|0.829|0.038|
> | CogVideoX VAE     |1.77|30.54|0.873|0.046|
> | HunyuanVideo VAE  |0.61|31.54|0.892|0.025|
> | IV VAE            |1.12|30.22|0.868|0.039|
> | Wan 2.1 VAE       |0.84|29.62|0.856|0.036|
> | Wan 2.2 VAE       |0.76|29.62|0.850|0.034|
> | DC-AE-V f32t4c256 |1.54|31.11|0.895|0.041|
>
> > _**Q2.1: Performance of DC-AE-V**_
>
> The original Video DC-AE is a non-causal autoencoder trained on 32 frames. It utilizes tiling and blending to process longer sequences, which creates a train-test gap during long video reconstruction. Specifically, we observe blurry reconstructions at the overlapped frames between two tiles, as shown in the supplementary material (e.g., frames 24 to 31 in temporal_modeling/opensora2ae_1.mp4). In contrast, the consistent training and inference of the chunk-causal approach in DC-AE-V results in smooth generalization to longer videos.
>
> > _**Q2.2: Novelty of DC-AE-V**_
>
> The key contribution of this paper is the introduction of the first successful framework for accelerating pretrained video diffusion models by adapting them to a deeply compressed latent space. This framework significantly boosts efficiency while preserving the base model's performance.
>
> Within this framework, DC-AE-V aims to push the upper limit of reconstruction for video autoencoders under deep compression. We achieve this by studying how to better exploit temporal redundancy within videos, which led to our chunk-causal solution.
>
> IV-VAE can be viewed as a special case of our chunk-causal design. However, IV-VAE strictly ties the group size to the temporal compression ratio, offering limited reconstruction accuracy gain over the standard causal design, as shown in Table 1 and Figure 5. Furthermore, DC-AE-V avoids the train-test gap observed in Video DC-AE when generalizing to longer videos.
>
> > _**Q3: AE-Adapt-V Baseline**_
>
> Thanks for your suggestions. The mentioned baseline was previously included in the appendix. We moved it to the main paper in the updated manuscript (Sec. 4.4 and Figure 9). Without a separate embedding alignment regression training, directly fixing the DiT weight and training the patch embedder and output head with diffusion loss still results in poor initialization, which causes instability in the later end-to-end LoRA fine-tuning.
>
> > _**Q4: DC-VideoGen Performance**_
>
> To address the concern about the training dataset, we also use the same training dataset to fine-tune the base model (Wan-2.1-T2V-14B and Wan-2.1-I2V-14B) without changing the latent space, and the performance is reported below. The performance does not differ significantly from the pre-trained weights, and this demonstrates that the dataset is not specifically curated for the Vbench score.
>
> Besides, we add human evaluation on Amazon Mechanical Turk in Section 4.5 of the updated manuscript, which also demonstrates the preference for DC-VideoGen over the Wan-2.1 base model.
>
> |Model|Overall Score|Quality Score|Semantic Score|
> |-|-|-|-|
> |Pretrained Wan-2.1-T2V-14B|83.73|85.77|75.58|
> |Fine-tuned Wan-2.1-T2V-14B|83.94|86.18|74.97|
>
> |Model|Overall Score|Quality Score|I2V Score|
> |-|-|-|-|
> |Pretrained Wan-2.1-I2V-14B|86.86|80.83|92.90|
> |Fine-tuned Wan-2.1-I2V-14B|86.72|81.19|92.25|

---

### Official Review · Reviewer_iJN1 · 2025-11-02

**Soundness:** 2
**Presentation:** 3
**Contribution:** 2
**Rating:** 4
**Confidence:** 4

**Summary:**

This paper introduces a post-training acceleration framework named DC-VideoGen, which aims to improve the inference efficiency and reduce the training cost of existing video diffusion models. The framework includes two main innovations: DC-AE-V and AE-Adapt-V. DC-AE-V is a video VAE that achieves 32x/64x spatial compression through a "chunk-causal temporal design," while maintaining high-quality reconstruction and generalization capability for longer videos. AE-Adapt-V is an efficient adaptation method that migrates a pre-trained diffusion model into the highly compressed latent space of DC-AE-V through "Video Embedding Space Alignment" and LoRA fine-tuning. Experimental results show that the framework reduce inference latency while maintaining or even slightly improving the base model's quality (VBench score), and it enables the generation of 2160p (4K) video on a single H100 GPU.

**Strengths:**

Improving the efficiency of the video generation models sounds and still needs exploration.

The writing is great and easy to follow.

**Weaknesses:**

Novelty and practical consideration of the temporal design (DC-AE-V): The paper's core 'chunk-causal' temporal modeling appears fundamentally similar to the Causal Group Convolution used in IV-VAE (CVPR'25). The authors claim the main advantage is a flexible chunk size, which they set to 40 for optimal results. This raises several concerns: (a) It is unclear how the model handles videos shorter than 40 frames. (b) For high-resolution generation, this large, fixed chunk size might lead to excessive VRAM consumption as it may not be divisible into smaller sub-chunks. (c) The claim of a dynamic chunk size being a core innovation is debatable, as a similar principle could seemingly be applied to IV-VAE, even if they only experimented with a size of 4.

Limited novelty at the framework level: the overall paradigm of this paper—training a VAE with a higher compression ratio and then adapting an existing model via post-training — is not entirely novel. For example, related work (like OpenSora 2.0) has already explored similar ideas. The innovation of this paper lies more at the component level (i.e., how to solve the specific technical challenges during adaptation).

**Questions:**

See the weakness for the questions.

---

> ### Author Response · Authors · 2025-11-24
>
> > _**Q1: DC-VideoGen Novelty and Contribution**_
>
> Our main contribution is a post-training framework designed to quickly adapt pre-trained video diffusion models to a deeply compressed latent space. This results in a large boost in efficiency while preserving the quality of the pre-trained model.
>
> Successfully adapting the autoencoder and retaining the base model's performance is non-trivial. Although OpenSora-2 has explored a similar idea, they mention that during adaptation, "video outputs appear blurry" ([1], Section 4.3), leading to the conclusion that "fast video generation model underperforms the original" ([1], Section 4.3). We also discovered a similar conclusion in experiments without AE-Adapt-V (Figures 6, 9 in our paper). However, with our AE-Adapt-V, we can perform a successful adaptation and reach the base model's performance.
>
> To further push the reconstruction upper limit of video autoencoders under deep compression settings, we additionally study how to better exploit temporal redundancy within videos. This leads to our chunk-causal solution. IV-VAE can be seen as a special case of our chunk-causal design, which strictly ties the group size to the temporal compression ratio, offering limited reconstruction accuracy gain over the standard causal design, as shown in Table 1 and Figure 5.
>
> > _**Q2: Additional Question about DC-AE-V**_
>
> - DC-AE-V on Shorter Videos: Modern video generation models (such as Wan, Hunyuan, and CogVideoX) primarily focus on long videos (80+ or 120+ frames), which leads to significant efficiency issues. DC-AE-V is specifically designed for these scenarios. To better support shorter videos (less than 40 frames), we suggest including such videos in the training dataset.
>
> - DC-AE-V Memory on High-Resolution Videos: Our DC-AE-V can directly encode and decode 720P video using a single H100. In low-memory settings, we can utilize spatial tiling to save memory.
>
>
>
> ---
> **Reference:**
>
> [1] Open-Sora Team, Open-Sora 2.0: Training a Commercial-Level Video Generation Model in $200k, https://arxiv.org/pdf/2503.09642

---

### Official Review · Reviewer_Ygub · 2025-11-02

**Soundness:** 2
**Presentation:** 3
**Contribution:** 2
**Rating:** 4
**Confidence:** 4

**Summary:**

This paper presents DC-VideoGen, a post-training acceleration framework designed to make existing video diffusion models more efficient. It relies on two key innovations: 1) a Deep Compression Video Autoencoder (DC-AE-V) featuring a novel "chunk-causal" design, which achieves high spatial compression (up to 64x) while preserving reconstruction quality; and 2) AE-Adapt-V, an efficient strategy to rapidly and stably adapt pre-trained models to the new compressed latent space. Experiments show this framework delivers up to a 14.8x inference speedup and enables 4K video generation on a single GPU, all while maintaining the original model's quality.

**Strengths:**

1. The paper tackles a critical and urgent issue in the video generation field: the high cost of training and inference. This research direction is highly valuable for democratizing and advancing high-resolution video synthesis.
2. The paper proposes a reasonable two-pronged solution. First, the novel chunk-causal video autoencoder (DC-AE-V) offers a clever theoretical approach to balancing high compression with long-video generalization. Second, the AE-Adapt-V strategy specifically addresses the instability issues encountered when adapting pre-trained models to new latent spaces, and its two-stage alignment is a technically sound design.
3. The paper reports astonishing efficiency gains (e.g., up to 14.8x speedup) while maintaining comparable or even slightly better performance on the VBench benchmark. If fully verifiable, this work would have a significant impact on the field of video generation.

**Weaknesses:**

1. In Section 3.3.1 and Figure 6, the authors construct a baseline method by retraining the DiT while randomizing the embedder and head. This approach is clearly suboptimal. A more conventional and intuitive baseline would be to first freeze the DiT and fine-tune only the embedder and head using the diffusion loss. I believe this approach would not suffer from training collapse and could achieve results comparable to AE-ADAPT-V. Therefore, the necessity and novelty of AE-ADAPT-V are questionable.
2. For video VAE reconstruction performance, the trend causal < chunk-causal < non-causal is clear. My doubt is whether the transition from causal to non-causal can yield a PSNR improvement of over 4 dB (as shown in Figure 5). In Table 1, increasing the number of channels by 4x results in only a 4.5 dB PSNR increase. In other words, the performance gain from causal to non-causal is claimed to be equivalent to a 4x increase in channel count, which seems implausible. In my past experience, the improvement from causal to non-causal is only around 2 PSNR points. In other papers that adopt a non-causal design, such as OpenSora 2 and VideoVAEPlus, the PSNR improvement over causal models like Hunyuan or CogVideoX is also less than 2 dB. Therefore, I have significant doubts about the upper limit of the performance gain from the chunk-causal design.
3. Could the authors provide a comparison with other methods on shorter frame counts (e.g., 40 frames)? The authors attribute the poor performance of non-causal models to their difficulty in generalizing to long videos, and their tests are conducted on 80 frames. When the test frame count is 40, the proposed method degenerates into a non-causal VAE. In this scenario, does the method still hold an advantage over Video DC-AE and VideoVAEPlus? If so, why? My understanding is that non-causal VAEs lack good temporal generalization, and their performance may drop significantly if the test and training frame counts do not match (e.g., Video DC-AE was trained on 32 frames, VideoVAEPlus on 16 frames). However, the authors' test frame count matches their own training frame count, which could introduce unfairness.
4. The VBench metrics reported in the paper are outstanding and astonishing. In Table 2, using DC-AE-V-f64t4c128 as the VAE, the model achieves a comparable VBench score with only 1/22 of the latency of Wan-2.1-1.3B. Since the paper does not provide model weights, I cannot verify the authenticity of this result. Therefore, I would like to know if the authors plan to open-source the weights of both the VAE and the generation models for the community to reproduce. If so, could they provide a reliable, latest possible release date?

**Questions:**

Please refer to weakness part.

---

> ### Author Response · Authors · 2025-11-24
>
> > _**Q1: AE-Adapt-V Baseline**_
>
> The mentioned baseline was previously included in the appendix. We moved it to the main paper in the updated manuscript (Sec. 4.4 and Figure 9). Two baselines, (1) the randomly initialized patch embedder and output head, and (2) joint fine-tuning of the patch embedder and output head with diffusion loss, both encounter training instability. Our AE-Adapt-V, in contrast, demonstrates stable training and superior performance during autoencoder adaptation, finally reaching the base model's performance.
>
> > _**Q2: Chunk-Causal Performance Gain**_
>
> The substantial performance gain in DC-AE-V might conflict with your prior knowledge, primarily due to misaligned settings. We mainly focus on the deep compression setting with a 32x spatial compression ratio. Therefore, in this setting, a larger chunk size will better exploit temporal redundancy and result in a larger performance gain.
>
> Besides, differences in implementation details also contribute significantly to the performance gap between non-causal (chunk-causal) and causal methods.
>
> - For non-causal (chunk-causal) methods, our DC-AE-V uses chunk size 40, larger than the training frames of Video DC-AE (OpenSora 2) and VideoVAEPlus, which are 32 and 16, respectively. A larger chunk size typically yields a performance gain, as validated in Figure 5. Therefore, our DC-AE-V would achieve better performance than these non-causal methods.
> - For causal methods, previous works like Hunyuan and CogVideoX pad 3 frames before the first frame, resulting in a larger latent size. To ensure a fair comparison, we do not pad frames for all the models in Figure 5, including the causal baseline. The following table shows that a causal autoencoder with 3 pad frames has better reconstruction accuracy than one without pad frames. Therefore, our causal baseline would perform worse than previous causal works.
>
> |  Implementation     |PSNR |
> |---------------|-----|
> |w/ 3 pad frames|29.73|
> |w/o pad frames |29.27|
>
> Given the combination of setting and implementation differences, we would expect a larger gap between our chunk-causal method and the causal baseline.
>
> > _**Q3: Video Autoencoder Reconstruction Comparison on Shorter Videos**_
>
> We add a reconstruction benchmark on 40-frame videos in Table 6 of the updated manuscript. We still observe an advantage over Video DC-AE and VideoVAEPlus. This is because Video DC-AE and VideoVAEPlus are trained on 16 or 32 frames, while ours DC-AE-V is trained on a chunk size of 40. As shown in Figure 5, increasing the chunk size from 16 or 32 to 40 still leads to a performance gain by fully utilizing temporal redundancy.
>
> > _**Q4: Reproducibility**_
>
> Thanks for your interest. The code and trained model will be released upon paper acceptance.

---

### Official Review · Reviewer_5X8c · 2025-11-02

**Soundness:** 3
**Presentation:** 2
**Contribution:** 2
**Rating:** 6
**Confidence:** 5

**Summary:**

The paper's central thesis is that massive, pre-trained video diffusion models can be made dramatically faster with negligible quality loss by forcing them to operate in a much smaller latent space. To achieve this, the authors first build a new, highly aggressive video autoencoder (DC-AE-V) and then propose a specific fine-tuning recipe (AE-Adapt-V) to port the original diffusion model over. The core of this recipe is an explicit alignment stage meant to prevent the model from collapsing during this transfer. The paper then presents a suite of VBench scores across various settings to argue that this aggressive compression strategy works as claimed.

**Strengths:**

The proposed chunk-causal mechanism is a specific and novel architectural contribution for video autoencoders.
The explicit embedding space alignment stage in AE-Adapt-V is a technically sound solution for the latent space mismatch problem.
The work provides a strong empirical demonstration of achieving extreme (e.g., 64x) spatial compression in a video generation pipeline.

**Weaknesses:**

1.	Chunk-causal is one of the paper's contributions, but its implementation is not well-described in the main paper. I initially assumed that the proposed chunk-causal design was implemented via attention masks. It was not until I reviewed the appendix, which shows the VAE employs ResNet blocks, that I came to suspect the process is similar to IV-VAE. Implementing chunk-causal with convolutions is not a conventional approach, and the authors are advised to describe this in the main text.

2.	As seen in Figure 9, the authors delay the temporal downsampling in the VAE and use a very large number of channels. This would lead to very high computational and parameter costs. The authors should provide a comparison with other VAEs in this regard.

3.	I noticed that the reconstruction results for Video DC-AE (with tiling) in the appendix show severe degradation before the 32nd frame. This is quite peculiar, as Video DC-AE itself was trained on 32 frames, so there is no reason for it to exhibit such poor reconstruction quality between frames 24 and 32. I suggest that the authors check the correctness of their relevant testing code to ensure the fairness of the results.

4.	The novelty of this paper is limited. For instance, AE-ADAPT-V combined with LoRA fine-tuning seems like a conventional approach. Chunk-causal appears to be an improvement based on IV-VAE, and while its reported gains are significant, I am not convinced it is truly that effective. In fact, after reading the paper, I attempted to reproduce the work, but my preliminary results suggest it is far less effective than reported, particularly in terms of SSIM and LPIPS metrics.

**Questions:**

See weaknesses

---

> ### Author Response · Authors · 2025-11-24
>
> > _**Q1: Chunk-Causal Implementation Details**_
>
> We add the implementation details in Algorithm 1 and lines 962–967 of the updated manuscript.
>
> > _**Q2: Efficiency Comparison of Video Autoencoders**_
>
> The Video Autoencoder (Video AE) takes a relatively small fraction of the total computation compared to the diffusion process. For example, when measuring the latency of 720P video generation using DC-VideoGen-Wan-T2V-14B on a single NVIDIA H100 GPU, the Video AE decoder's latency accounts for only 3.49% of the total time (as shown below). Additionally, we have included a Video AE efficiency benchmark in Table 8 of the updated manuscript. DC-AE-V demonstrates comparable decoding latency to the previous Wan-VAE or Video DC-AE.
>
> | Component | Latency (s/video) | Percentage |
> | :--- | :--- | :--- |
> | Diffusion Process | 206.41 | 96.51% |
> | Video AE Decoder | 7.46 | 3.49% |
>
> > _**Q3: Video DC-AE Reconstruction Results**_
>
> It is expected to observe worse reconstruction results for Video DC-AE between frames 24 and 32. We attribute this issue to overlapping and blending between the two tiles. The official implementation of Video DC-AE sets the overlap ratio to 0.25, resulting in 8 overlapped frames between two adjacent 32-frame tiles. For example, the first tile spans from frame 0 to frame 31, the second tile spans from frame 24 to frame 55, and frame 24 to 31 are the overlapped ones. The encoding and decoding outputs of these overlapped frames are blended from the two tiles. This blending leads to a gap between training and inference, resulting in blurry reconstruction results.
>
> > _**Q4: Contribution and Novelty of DC-VideoGen**_
>
> Our main contribution is a post-training framework designed to quickly adapt pre-trained video diffusion models to a deeply compressed latent space. This results in a large boost in efficiency while preserving the quality of the pre-trained model.
>
> Within this framework, AE-Adapt-V is the first successful solution that solves the autoencoder adaptation problem for video diffusion models. The key design behind AE-Adapt-V is the introduction of embedding space alignment. Without this design, simply using LoRA finetuning would lead to instability and a significant performance drop, as illustrated in Section 4.4 and Figure 9.
>
> Furthermore, the performance gain of the proposed DC-AE-V over IV-VAE is traceable. IV-VAE can be seen as a special case of our chunk-causal design, which strictly ties the chunk size to the temporal compression ratio (e.g., 4). In contrast, DC-AE-V decouples these two factors and can benefit from a larger chunk size (e.g., 40), as shown in Figure 5.
>
> > _**Q5: Reproducibility**_
>
> Thanks for your interest. The code and trained model will be released upon paper acceptance.

---

### Author Response · Authors · 2025-11-24

We sincerely thank all reviewers for their constructive comments. We are encouraged to see that the reviewers recognize our contributions to video generation acceleration. We have updated our manuscript accordingly (highlighted in gold color). Please see our detailed responses below each review. We greatly appreciate your feedback and welcome any additional questions.

---

### Meta-Review · Area_Chair_7mCs · 2026-01-03

**Summary:**

The paper presents a new deep-compression video autoencoder based on chunk-casual encoding. Moreover the paper presents a new adaptation method for fine-tuning existing generation model for new autoencoder.

The paper presents a rather ambitious claim, that video generator with high compression autoencoder can be competitive with video generator with regular autoencoder. Based on area chair own experience, high compression AE usually comes at the cost of increased number of low level artifacts and overall lower level of details, which is clearly can be seen in visuals, likely cherry-picked results, from supplementary material. This issue is also highlighted by reviewer 3xJk.

As also noted by 3xJk and Ygub, VBench scores looks very high for this paper, while qualitative results does not support them. It is well know that VBench scores, have large variance and usually is not very indicative of the model performance. During the discussion phase, authors provide an additional user study, in which proposed method preferred significant amount of times over Wan 2.1, which is both not explained and not intuitive.

Overall given weak qualitative results and not well addressed questions regarding evaluation, area chair recommends a rejection.

**Reviewer Concerns:**

5X8c, iJN1, 3xJk - have novelty concerns. These where not address, however are chair admits that these concerns are in general impossible to address. Overall these concerns was not the main factor in the paper decision.

5X8c - raised concern about parameter cost of the proposed auto-encoder which was not addressed.

5X8c - also raised concerns about quality degradation for the specific frames for one of the baselines, which may indicate a bug in implementation. The authors provide an explanation of why degradation may happen, however, after reply it was still unclear if the inference was performed correctly.

Ygub and 3xJk - believe fine-tuning method to be too complicated and also believe that simpler baselines can be much better. Author claim that simpler baselines may lead to unstable training. This contradict area chair experience, usually adapting to new AE space works with any simple method, initializing heads from random or first training heads, given that hyperparameters selected correctly. So this concern is not very well addressed according to area chair.

Finally the main concern that was raised by Ygub and 3xJk. VBench metrics that was is aligning with perceptual judgement of 3xJk and are chair. See details above.

**Reviewer Scores:**

5X8c - stay the same
Ygub - stay the same
iJN1 - stay the same
3xJk - stay the same

---

### Decision · Program_Chairs · 2026-01-26

Reject